# OpenReview forum: "Can Language Models be Instructed to Protect Personal Information?"
_ICLR.cc/2024/Conference — Submitted to ICLR 2024_

### Official Review · Reviewer_unBh · 2023-10-31

**Soundness:** 3 good
**Presentation:** 3 good
**Contribution:** 3 good
**Rating:** 6
**Confidence:** 4

**Summary:**

==*== Paper summary

This paper discusses the impact of large multimodal language models in various applications and the privacy and security concerns associated with their use. These models have been observed to memorize and unintentionally disclose pre-training data, posing significant privacy risks. The paper introduces PRIVQA, a multimodal benchmark designed to evaluate the trade-off between privacy protection and model utility. PRIVQA simulates scenarios where models are instructed to safeguard specific categories of personal information. The evaluation on PRIVQA assesses how well access control instructions can prevent models from leaking protected personal data. The paper also presents a method for iteratively moderating model responses, enhancing privacy. However, the study reveals that adversaries can easily bypass these protections through simple jailbreaking techniques using text or image inputs. The authors suggest that PRIVQA can aid in the development of improved privacy protection in models and enhance their resilience against adversarial attacks.

**Strengths:**

==*== Strengths

+ The paper proposes PRIVQA, a multimodal benchmark designed to evaluate the trade-off between privacy protection and model utility.
+ The research question is well defined and valuable to the research community.
+ Extensive case studies.

**Weaknesses:**

==*== Weaknesses

- The convincingness of the output experimental results still needs to be further improved.
- Comparisons with more advanced baseline methods are needed to highlight the advantages of the proposed privacy preserving techniques.
- The technical depth of this paper needs to be further improved.

**Questions:**

==*== Comments for author

Q1: In Figure 9, the authors present an illustrative case to expound upon the privacy-utility trade-off inherent in the current GPT-4 model, particularly with respect to its handling of location information. Evidently, the showcased examples elucidate the inadvertent disclosure of image location data by GPT-4. Nevertheless, a pertinent query arises as to whether GPT-4 would also inadvertently divulge location information in the case of images depicting unfamiliar landmarks or attractions. To address this question comprehensively, it is imperative to consider the pervasive practice of mobile devices embedding geospatial data within the photographs they capture. In the event that an adversarial agent can effectively employ tailored adversarial prompts to prompt GPT-4 to discern the geographic origin of such photographs, it would substantially enhance the persuasiveness of the assertion.

Q2: More baselines still need to be added to illustrate the superiority of the proposed self-moderation technology. Indeed, the reviewer knows that the main contribution of this paper is not the designed self-moderation technology, but it would be better if existing privacy protection technologies could be more comprehensively explored to illustrate the privacy concerns of existing multi-modal large language models. For instance, it would be judicious to incorporate emerging data right-to-forget protection technologies, such as machine unlearning, as a baseline reference. By doing so, the authors can provide a more robust and convincing assessment of the available privacy-preserving techniques within the context of multi-modal large language models.

Q3: The reviewers acknowledge that the PRIVQA benchmark data set proposed in this paper already includes a variety of privacy attribute information. I might expect that the benchmark data set could include more sensitive private information such as race, occupation, address, bank account, etc.

Q4: To be candid, I find myself in a somewhat uncertain position regarding the alignment of the proposed benchmark dataset with the prevailing perspectives within the ICLR community. There exists a degree of ambiguity concerning whether the technical intricacies involved in dataset creation adequately conform to the community's overarching scope and the stipulated requirements for paper submissions. It may be prudent to consider that the NeurIPS dataset track could potentially represent a more suitable forum for the presentation and evaluation of this dataset, given its specific focus and expertise in dataset-related matters.

---

> ### Author Response · Authors · 2023-11-13
>
> We thank the reviewer for their thoughtful review. We have responded to your questions below:
>
> **(Q1) Using geospatial location metadata as inputs to vision language models**
>
> We do not use location metadata as inputs to vision language models, rather the inputs are simply the query and the image. However, this is an interesting question for future work, but we see this as beyond the scope of the current paper.
>
> **(Q2) Comparisons with additional baseline methods are needed to highlight the advantages of the proposed privacy-preserving techniques.**
>
> Thank you for this suggestion! We have added a simple baseline confidence method based loosely on a technique from (Portillo Wightman et al., 2023): https://aclanthology.org/2023.trustnlp-1.28/.  A figure containing the results of this method for llama2-chat-7B is shown in Appendix B (see Figure 17).
>
> You have also mentioned adding a machine unlearning baseline. In Section 4 (see Figure 6), we include a model editing baseline using the ROME (Meng et al., 2022) methodology, through which we find that this technique struggles to scale to the many model edits required to adhere to access control instructions. We choose to include a model editing baseline instead of a machine unlearning baseline as it is more pertinent to our work. Specifically, machine unlearning attempts to ‘forget’ certain sequences of training data, whereas our work is concerned with preventing the disclosure of information (not necessarily verbatim from the training corpus) that is protected. This approach is more similar to model editing work such as ROME which focuses on editing factual associations in language models.
>
> **(Q3) The benchmark data set could include more sensitive private information such as race, occupation, address, bank account, etc.**
>
> One of the reasons that we chose to construct our dataset based on public figures is to avoid including sensitive information about private figures. Please see the added ‘Ethics Considerations and Limitations Section’ for additional information. However, we do include some of the attributes you mentioned in our dataset such as citizenship and occupation as these are categories present in Wikidata. We also find that models do exhibit biases with respect to race (see Figure 20).
>
> **(Q4) Alignment of the proposed benchmark dataset with the prevailing perspectives within the ICLR community**
>
> Task definition/evaluation papers have previously been well accepted at ICLR. For example, from last year’s conference, the following two papers are related in style to ours:
>
> Shi et al., Language models are multilingual chain-of-thought reasoners: https://openreview.net/forum?id=fR3wGCk-IXp
>
> Tamkin et al., Task Ambiguity in Humans and Language Models: https://openreview.net/forum?id=QrnDe_9ZFd8

---

> > ### Comment · Reviewer_unBh · 2023-11-21
> > **Response to Authors**
> >
> > Thanks to the authors for their detailed and timely responses! First, the authors have addressed most of my concerns. However, I still have the following questions that need further answers:
> >
> > - Regarding privacy attacks, I am curious whether the method used in the article can infer the location information of any image?
> > - Regarding the practicality of the attack, I am curious if LLM leaks personal information, how can the adversary use this information to carry out more harmful actions in the next step?
> >
> > I look forward to responses from the authors! At least for now, I still maintain my score.

---

> ### Author Response · Authors · 2023-11-22
>
> Thank you for your response and for these important questions! We have answered them below:
> * Geolocation will not work for every image, with the success being highly dependent on the landmarks in the image. We ran a small human evaluation of model outputs (IDEFICS-80B and GPT-4V) on a random sample of 32 images from the InfoSeek dataset and found that **53.1% (17/32)** of images were geolocated successfully by GPT-4V and **31.3% (10/32)** of images were geolocated successfully by IDEFICS-80B. With recent improvements in model capabilities, we expect this number to increase and believe that it constitutes a real risk to user privacy. Due to the limited time before the end of the author response period, we have not updated the paper with these results but will do so for the camera-ready version of the paper.
> * There are a number of potential scenarios in which the adversary can use leaked personal information from LLMs to carry out harmful actions in both the digital and physical worlds. We have outlined some of these below and will also update our paper with a discussion of these:
> 1. **Aiding in spear phishing attacks (https://en.wikipedia.org/wiki/Phishing):** An adversary can collect personal details about an individual (e.g., their place of work, location, etc.) to help them craft compelling emails to be used in spear phishing attacks. Note that even if a user deleted this type of information from their social media, the model will still be able to divulge their information if it was initially included in the pretraining corpus.
> 2. **Aiding in doxxing attacks (https://en.wikipedia.org/wiki/Doxing):** A vision language model could be used for the identification of a normally private individual from a photo or video  (potentially a viral video circulating on social media) enabling malicious users to harass the now-identified individual on their social media accounts.
> 3. **Aiding in physical stalking (https://en.wikipedia.org/wiki/Stalking):** Identifying private information about an individual, especially their current location (e.g. based on photos they recently posted on social media) using a vision language model, to enable physical stalking.
>
> We are happy to follow up on any of these questions or respond to any other comments you have before the end of the response period on November 22nd (tomorrow).

---

> > ### Comment · Reviewer_unBh · 2023-11-22
> > **Response to Authors**
> >
> > Thanks to the authors for their prompt replies! The authors have addressed some of my concerns, but regarding image privacy I still think that the existing evidence is not enough to prove that GPT-4 has serious privacy leaks. I look forward to seeing more experimental results. Therefore, I still maintain my score.

---

### Official Review · Reviewer_yBLS · 2023-10-31

**Soundness:** 3 good
**Presentation:** 4 excellent
**Contribution:** 4 excellent
**Rating:** 6
**Confidence:** 4

**Summary:**

The paper introduces a new benchmark PrivQA to study the ability of language model to follow instructions about access control and to study the privacy utility trade-off. PrivQA includes both textual and multimodal tasks, where the models are asked to answer questions or abstain from answering if they involve sensitive data. The paper evaluates current generation models such as GPT-4 and Llama-2. The authors find that these models leak a lot of private information even if they're instructed not to. Self moderation improves the results significantly. The authors also consider an adversarial settings and find that under a multi-hop threat model attacks succeed nearly always.

**Strengths:**

- The paper provides a valuable benchmark for privacy protection in language models, which is an emerging and important research area. There are few existing datasets that focus on privacy issues in language models.
- The paper uses state-of-the-art models for the evaluation, which makes the results more relevant and convincing.

**Weaknesses:**

- The paper does not share the code or data to reproduce the results, which limits the reproducibility and verifiability of the work. The paper says the URL is removed for review, but there are ways to share it anonymously (e.g. anonymous.4open.science).
- The paper uses evaluation metrics that do not capture the severity of privacy breaches. Privacy is about preventing the worst-case scenarios, not the average ones. Therefore, privacy metrics should reflect that even a single leak of private data is unacceptable. For example, in differential privacy, δ is set to a very small value or in membership inference attacks, TPR is reported at very low FPR. The paper shows some trends of privacy improvement (e.g. self moderation), but none of the methods offer adequate protection for realistic scenarios.

**Questions:**

- Clarification in section 3.1. Is it true that $\mathbb{P}\mathbb{G}\cup\mathbb{C}\mathbb{G}=\mathcal{X}$? If so, it might be helpful to state this rather than the subset relation.
- Why are there such noticable differences in model sizes in figure 2 and why do larger models have more privacy violations? I would expect that larger models follow instructions better.
- Clarification: What are head entities as described in the last paragraph on page 7.
- Could you respond to the 2nd point in weaknesses? Would you consider any of the presented methods adequate for use in a realistic setting?

---

> ### Author Response · Authors · 2023-11-13
> **Author Response**
>
> We thank the reviewer for their insightful feedback. Responses to each of your points are listed below:
>
> **Open-sourcing code and data**
>
> To facilitate the reproducibility of the research work, we uploaded data/code to the anonymized link here: [link](https://anonymous.4open.science/r/submission-iclr-5AC7/README.md). Due to the size of the data (2.4GB), we uploaded them to Google Drive using an anonymous email (submissionanonymized@gmail.com).
>
> **Privacy is about even a single leak of private data is unacceptable.. would you consider any of the presented methods adequate for use in a realistic setting?**
>
> We agree that leaking private data is unacceptable, and we have been very careful to avoid claiming that current models are good enough at this. We pointed out the serious issue in the introduction as
> ```
> However, we discover serious issues related to bias and robustness with these instructions that we believe need to be addressed before they can be used in critical applications.
> ```
> In fact, we hope our experimental results (non-adversarial and adversarial) emphasize this point and alert practitioners to be aware of the limitations of current models.
>
> **Clarification: section 3.1, is it true that $\mathbb{PG} \cup \mathbb{CG} = \mathcal{X}$**
>
> Thanks! This is true and we have revised the manuscript to make this clear.
>
> **Clarification: (Figure 2) Why do larger models have more privacy violations? I expect larger models to follow instructions better**
>
> Indeed, we show the **protection score** of llama-70b outperforms llama2-chat-7b in Figure 2 (left: 75 vs 58, right: 72 vs 44).
> We suspect R3 was referring to the charts over *Response F1 (Protected)*, where the F1 score decreases as the model size increases from llama2-chat-7b to GPT-4. We would like to clarify this score should be as low as possible to make sure models don't reveal private information during response generation. Hence, the larger the model, the lower the score, which is expected as R3 mentioned.
> We will again clearly state in the revised manuscript to remove any confusion.
>
> **Clarification: what are head entities on page 7**
>
> * Head entities: popular entities that are more regularly visited on Wikipedia (e.g., $10^6$ Wiki pageviews).
> * Tail entities: less popular entities at the tail of the distribution (e.g., $10^1$ Wiki pageviews).
> * This is discussed in paragraph 8 of section 4.

---

> > ### Author Response · Authors · 2023-11-22
> > **Response to yBLS**
> >
> > Thank you again for your questions and thoughtful feedback to improve the paper. We have attempted to address your concerns in the response above and in the updated version of the paper. We would appreciate it if you could look over our response and update your score if we have adequately addressed your concerns. We're also happy to address any additional concerns you might have before the end of the discussion period on November 22 (tomorrow).

---

> > ### Comment · Reviewer_yBLS · 2023-11-22
> >
> > Thank you for your answering my questions, providing clarification and uploading the dataset.
> >
> > I still have concerns about the chosen metric (section 3.3 protection score). I think it doesn't clearly convey what the privacy risk is. Providing the specificity separately (or perhaps even clearer 1-specificity) would allow practitioners to understand how much leakage there is.

---

> ### Author Response · Authors · 2023-11-22
> **Response to yBLS**
>
> Thank you again for your response and feedback!
> We agree and plan to specifically report the two metrics ('specificity' and 'sensitivity') that are used to compute the protection score (harmonic mean) in the camera-ready version (due to limited time). Below we report these metrics for GPT-4 & llama2-70b.
>
> The reason we chose to report the protection score was mainly because of the motivation to prevent the model from
> over-optimizing for just one of the measures:
> In some extreme cases, one model can:
> - always answer the question --> 100% specificity and 0% sensitivity
> - always abstain to answer --> 0% specificity and 100% sensitivity
>
> But we appreciate the reviewer bringing this to our attention and we plan to report these two metrics to provide more context.
>
>
> Protected population:
> |method|model|sensitivity|specificity|protection score|
> |---|---|---|---|---|
> Self-moderation | GPT-4 | 80.7 | 92.3 | 86.1
> Self-moderation | llama2-70b | 72.8 | 82.1 | 76.0
>
> Protected information:
> |method|model|sensitivity|specificity|protection score|
> |---|---|---|---|---|
> Self-moderation | GPT-4 | 96.2 | 88.7 | 91.5
> Self-moderation | llama2-70b | 71.9 | 75.6 | 73.2

---

### Official Review · Reviewer_VxoZ · 2023-11-01

**Soundness:** 3 good
**Presentation:** 3 good
**Contribution:** 1 poor
**Rating:** 5
**Confidence:** 4

**Summary:**

This paper introduces a new dataset for evaluating how well instruction-induced privacy protection mechanisms work. They also propose a mitigation over just using instructions, and then red-team the proposed mitigation.

The proposed dataset is an augmentation of 5 existing datasets, where protected groups and information is defined over the data, and the goal is to get utility of the QA task, without revealing the sensitive information. The tasks are defined both for open-ended generation and for visualQA. The desired behavior is that if the model is asked a question regarding a sensitive topic, it abstains from answering, but answers utility related questions. The proposed mitigation is self-moderation, where the model is given its own response and asked to improve it.

The authors then red-team the proposed method through prompt injection and show that more fundamental solutions are necessary.

**Strengths:**

1. The paper looks at an important problem, as LLMs are bing used more and more and prompting and instruction tuning is ubiquitous.

2. I particularly like the visual aspect of the work and looking into multimodal models, as there aren't many existing works that focus on these models.

**Weaknesses:**

1. The threat model of the paper is not at all clear, neither is the paper positioned well among prior work. What is the privacy definition? What are we trying to protect, is it training data?  inference data? what is the actual application that the authors are targeting? what is the real world use-case?  It seems like the authors are targeting training data, however, according to existing extraction attacks [1], this is not a realistic scenario and not a real problem. There is no successful extraction attack that would so easily, from an instruction tuned model, extract any information that is not repeated many many times. The questions work in this dataset, since the authors target celebrities. So I think trying to protect training data here is not really sensible. The inference data, that would make more sense, which seems to be the case in the visual part of the paper? But then, in that case, what is a realistic scenario? In general it is very unclear what is happening here.

2. The dataset is not curated in a principled way and is too artificial. There are no levels to what is being protected, and how it needs protection. The attributes are artificially inserted.

3. I think the authors need to first discuss the related work better: talk about existing membership inference and extraction attacks, about the risks, and then about differential privacy and other protection methods, and then position the paper with regards to all that.

[1] Carlini N, Ippolito D, Jagielski M, Lee K, Tramer F, Zhang C. Quantifying memorization across neural language models. arXiv preprint arXiv:2202.07646. 2022 Feb 15.

**Questions:**

See above.

---

> ### Author Response · Authors · 2023-11-13
> **Author Response**
>
> We thank the reviewer for their comments to help us improve the paper. Responses to each of your points are listed below:
>
>
> **What is the threat model?  Is the data we are trying to protect from the training data, or data provided to the model at inference time? I think the authors need to first discuss the related work better.**
>
> This is an excellent suggestion. We agree that there should be a clearer discussion of the differentiation between our approach and related work. We have modified the introduction to provide this context. Specifically, while related to data extraction attacks and mitigation measures like machine unlearning, our setting and approach focus on more realistic and powerful privacy controls. For instance, while techniques like machine unlearning and model editing can be applied to edit or remove a specific text sequence or a single association found in the pretraining corpus such as “the name of the valedictorian of Central High School”, they cannot be easily used on a more realistic use case involving a category of information such as “the data of all students at Central High School”. We consider these more realistic privacy controls that enable adherence to these real-world privacy protection requirements (e.g., refusing to answer questions about a group defined with natural language).
>
>
> **What is the real-world use case?  It seems protecting information about public figures is not a realistic scenario?**
>
> This is a good point! We chose to include questions about public figures that have an associated Wikipedia page primarily because of ethical considerations associated with a widely distributed benchmark. A benchmark evaluating the ability to extract data for less well-known individuals would not be able to be released publicly, and could not serve as a standardized benchmark, although we agree that this more realistic benchmark would also have value.  This is discussed briefly in the first paragraph of Section 3, but we agree this is an important point and could be highlighted more clearly, so we have a new section “Limitations and Ethical Considerations” to highlight this discussion.
>
> **There is no successful extraction attack that would so easily, from an instruction tuned model, extract any information that is not repeated many many times.**
>
> We agree that data probably needs to appear multiple times in the training corpus for successful extraction. However, there are real-world cases where non-public entities might have their data appear multiple times in the corpus. For instance, we found that it was possible to extract personal information (the birthday of a user) about a Reddit user using the BetterDan jailbreaking prompt with gpt-3.5-turbo (an anonymized example has been added - see Figure 13) . Again, due to data restrictions and ethical concerns, we cannot publish a dataset of such examples, but we believe that PrivQA is a good alternative that has the potential to support widespread benchmarking and reproducible experiments.
>
> We also believe that answering questions about personal information related to public figures is a more practical way to evaluate LLMs’ capabilities to selectively protect personal information, as they are more likely to be able to answer these questions.  Accurately evaluating the ability of models to selectively refuse to answer in a scenario where the information appears only a few times is more challenging, as there will be higher variance in models’ ability to answer these questions.  We agree this is an important problem for future work to address, but we also feel that our PrivQA benchmark is a first step that is complementary to a future more realistic data-extraction benchmark, which would require more restrictive access to researchers and would have larger variance in the evaluation of models’ ability to selectively protect information due to higher difficulty in answering the questions.
>
> **The dataset is not curated in a principled way and is too artificial. There are no levels to what is being protected, and how it needs protection. The attributes are artificially inserted.**
>
> Thank you for the feedback. We agree that the motivations behind some of the protection attributes chosen could be better explained, and modified the original motivations presented in Section 3.1 in the Protected Populations and Protected Information paragraphs. The attributes used to define protected classes are taken from common WikiData categories and represent what we believe to be important privacy considerations based on Article 4 of the General Data Protection Regulation (https://gdpr-info.eu/art-4-gdpr/). For instance, it is likely that LLM developers will want to avoid disclosing information about minors, which motivates the use of attributes such as age and protected classes such as individuals under the age of 18.

---

> > ### Comment · Reviewer_VxoZ · 2023-11-22
> > **rebuttal ack**
> >
> > I want to thank the authors for their responses. The main thing I am still not fully clear on/convinces about, is the threat model. I know understand that this is supposed to be like machine unlearning, however, machine unlearning protects training data, but this approach seems more geared towards inference? I still don't see a direct real-world use-case of this.
> >
> > Also, while I appreciate the updates and modifications, I still think the paper needs more analysis so I am keeping my score for now.

---

> ### Author Response · Authors · 2023-11-13
> **Author Response**
>
> **Talk about existing membership inference and extraction attacks, about the risks, and then about differential privacy and other protection methods**
>
> Thank you for this suggestion! In the introduction and related work sections, we have included a clearer discussion about the relationship between our work and the related work you mention.
> We have also cited the following membership inference attack papers
>
> [1] Virat Shejwalkar, Huseyin A Inan, Amir Houmansadr, and Robert Sim. Membership inference attacks against nlp classification models. In NeurIPS 2021 Workshop Privacy in Machine Learning, 2021.
>
> [2] Abhyuday Jagannatha, Bhanu Pratap Singh Rawat, and Hong Yu. Membership inference attack susceptibility of clinical language models. arXiv preprint arXiv:2104.08305, 2021.
>
> [3] Fatemehsadat Mireshghallah, Kartik Goyal, Archit Uniyal, Taylor Berg-Kirkpatrick, and Reza Shokri.Quantifying privacy risks of masked language models using membership inference attacks. arXiv preprint arXiv:2203.03929, 2022.
>
> Please let us know if you would like to see any other particular papers cited.

---

> > ### Author Response · Authors · 2023-11-22
> > **Response to VxoZ**
> >
> > Thank you again for your questions and thoughtful feedback to improve the paper. We have attempted to address your concerns in the response above and in the updated version of the paper. We would appreciate it if you could look over our response and update your score if we have adequately addressed your concerns. We're also happy to address any additional concerns you might have before the end of the discussion period on November 22 (tomorrow).

---

### Official Review · Reviewer_YNv6 · 2023-11-01

**Soundness:** 2 fair
**Presentation:** 2 fair
**Contribution:** 2 fair
**Rating:** 3
**Confidence:** 4

**Summary:**

This paper presents PRIVQA, a multimodal benchmark designed to evaluate the balance between privacy and utility in scenarios where a model must safeguard certain types of personal information. The performance of language models is assessed using PRIVQA to determine their ability to adhere to access control instructions and avoid revealing sensitive personal data. Additionally, this paper introduces a method for models to self-moderate their responses in a way that greatly enhances privacy protection.

**Strengths:**

Pros:
1. This research presents an open benchmark designed to evaluate language and vision models on their ability to safeguard personal information by adhering to instructions.
2. The study introduces a self-moderation approach that enhances the proficiency of models in complying with access control directives, while also revealing persistent biases in the protection afforded to diverse groups.
3. The paper details a sequence of red-teaming exercises, highlighting that current advanced models can be readily bypassed by adversarial methods when following access control instructions.

**Weaknesses:**

Cons:
1. The technical novelty is limited. This paper just tests whether or not the conventional instruction-tuned LLMs can protect privacy. The proposed “Self-Moderation” seems to slightly modify the previous “reflection” techniques in many previous works (there is a survey [1] on “reflection” techniques).
2. The title is misleading. The title is not very related to the core message of this paper because this paper does not conduct instruction tuning to protect privacy but just test whether or not the conventional instruction-tuned LLMs can protect privacy. So, the title should not be “Can Language Models be Instructed to Protect Personal Information?” but “Can Instruction-tuned Language Models Protect Personal Information?"
3. The contribution in the read teaming part is unclear. It seems this paper just directly applies the previous red teaming methods for an empirical study and do not propose any new read teaming method.
4. The connection between the read teaming part in Section 5 and privacy experiments in Section 4 is unclear. Can I also regard the privacy experiments in Section 4 as “read teaming”? Because the authors define “Red teaming has become a standard method to elicit and evaluate privacy, security, and bias concerns with language models” in Section 5. Based on my understanding, the privacy experiments in Section 4 are also “read teaming”. It is unclear why the authors define “read teaming” in Section 5 again. The content in Section 4 & Section 5 is overlapped.
5. This paper does not discuss any effective ways to protect privacy. Although this paper conduct an assessment on LMMs to follow instructions to protect personal information. The proposed Self-Moderation strategy seems to be not very effective. It is suggested the authors provide more insights on how to effectively protect privacy.


[1] Automatically Correcting Large Language Models: Surveying the landscape of diverse self-correction strategies https://arxiv.org/abs/2308.03188

**Questions:**

See weaknesses.

---

> ### Author Response · Authors · 2023-11-15
> **Authors Response**
>
> We thank the reviewer for their comments. Responses to each of your points are listed below:
>
> **This paper just tests whether or not the conventional instruction-tuned LLMs can protect privacy.**
>
> Yes, we do test the extent to which instruction-tuned LLMs can protect privacy. However, we believe that this is a significant contribution because
> * our task formulation is more practical than other definitions that currently exist in the literature such as protecting against extracting specific sequences of training data. This new personal information protection task formulation can be a source of future work focused on developing better methods for this task.
> * we provide a forward-looking open-source benchmark dataset in the form of PrivQA that can be used for benchmark future LLMs on our defined task.
> * using our constructed dataset, we demonstrate the failure modes of existing LLMs specifically regarding bias and adversarial robustness, which can again serve as a basis for improvement for future models and protection techniques
>
>
> **The proposed “Self-Moderation” seems to slightly modify the previous “reflection” techniques in many previous works (there is a survey [1] on “reflection” techniques).**
>
> Thank you for the reference to this survey paper! We cited this survey in the updated version of the paper. We would like to preface this response by making it clear that we do not claim that Self-Moderation is a key contribution of our paper (as mentioned these would be the novel task definition and the benchmark dataset).
> We agree that our Self-Moderation technique is based on these “reflection” techniques and tried to make this clear originally by citing Self-refine: Iterative refinement with self-feedback (Madaan et al., 2023). However, like many of the papers in the survey paper, the main differentiation factor with our technique (Self-Moderation) is its application, which in our case is privacy and access control. Specifically, we show that LLMs perform somewhat well in the role of an access control system and that LLM decisions become increasingly conservative regarding data protection with more refinement.
>
>
> **The title is misleading. The title is not very related to the core message of this paper because this paper does not conduct instruction tuning to protect privacy but just test whether or not the conventional instruction-tuned LLMs can protect privacy.**
>
> In the paper, we do not claim that we perform instruction finetuning on LLMs to improve their ability to protect privacy, but instead evaluate the extent to which widely used LLMs can respond to personal information protection instructions.
> We believe that both our current title and the alternative title you supplied have similar meanings and reflect the claims that we make in the paper.
>
> **The contribution in the red teaming part is unclear. It seems this paper just directly applies the previous red teaming methods for an empirical study and do not propose any new red teaming method.**
>
> We agree that we do use previously proposed red teaming methods including standard jailbreaking prompts like BetterDan and AIM. These methods were included as baselines to study the robustness of access control instructions to measure their ability to be applied to potentially adversarial real-world scenarios. We do not claim that these are contributions.
>
> **However**, we do apply other more novel methods including visual prompt injection, or modifying input images to VLMs to induce targeted failure modes i.e. revealing personal information (see Figure 8). While there have been two works concurrent with ours that have studied visual prompt injection:
> * Not what you’ve signed up for: Compromising Real-World LLM-Integrated Applications with Indirect Prompt Injection (Greshake et al. 2023)
> * Plug and Pray: Exploiting off-the-shelf components of Multi-Modal Models (Shayegani et al., 2023)
> We believe we were the first to consider this attack for truly sensitive real-world applications.
>
>
> **The connection between the read teaming part in Section 5 and privacy experiments in Section 4 is unclear. The content in Section 4 & Section 5 is overlapped.**
>
> We apologize if this was unclear. We consider red teaming methods to employ adversarial inputs to circumvent access control instructions. The goal of Section 4 was to present a comprehensive evaluation of existing LLMs using our constructed benchmark. While this evaluation did include an analysis of shortcomings of access control instructions such as popularity protection biases, these were not intentionally induced. On the other hand, the goal of Section 5 was to present methods to adversarially circumvent access control instructions with malicious inputs.

---

> > ### Author Response · Authors · 2023-11-15
> > **Author Response**
> >
> > **This paper does not discuss any effective ways to protect privacy. Although this paper conduct an assessment on LMMs to follow instructions to protect personal information. The proposed Self-Moderation strategy seems to be not very effective.**
> >
> > We agree that for current LLMs the instruction-based methods that we present in this paper are not sufficient for sensitive applications that require high robustness as we mention in the introduction:
> > ```
> > However, we discover serious issues related to bias and robustness with these instructions that we believe need to be addressed before they can be used in critical applications.
> > ```
> > By pointing out limitations in current models and contributing a new task definition and benchmark, we hope that our work will facilitate the development of future models that perform better in this regard.

---

> > > ### Author Response · Authors · 2023-11-22
> > > **Response to YNv6**
> > >
> > > Thank you again for your questions and thoughtful feedback to improve the paper. We have attempted to address your concerns in the response above and in the updated version of the paper. We would appreciate it if you could look over our response and update your score if we have adequately addressed your concerns. We're also happy to address any additional concerns you might have before the end of the discussion period on November 22 (tomorrow).

---

### Author Response · Authors · 2023-11-13
**General Response to Reviewers**

We thank the reviewers for their valuable feedback and suggestions to improve the paper.

We are delighted to see that the reviewers found value in this work:
* “the paper looks at an important problem… I particularly like the visual aspect of multimodal models… as there aren't many existing works” [VxoZ]
* “the paper provides a valuable benchmark for privacy protection in language models, which is an emerging and important research area” [yBLS]
* “the research question is well defined and valuable to the research community” [unBh]

We have responded to the reviewers' comments individually and have updated the paper with the following high-level changes:
* As suggested by yBLS, we have added an anonymized code repository to allow for reproducibility during the response period: https://anonymous.4open.science/r/submission-iclr-5AC7/README.md
* As suggested by VxoZ, we have updated our introduction and related works section to better contextualize our contributions and task in terms of previous work.
* VxoZ and unBh had questions about private information in the dataset. Specifically, VxoZ wondered why we used public entities to construct our dataset instead of private ones. We believe that this is a very important question. The short answer is that this decision was made based on data restrictions and ethical considerations. We have added an ‘Ethical Considerations and Limitations’ section in the main text to more fully answer this question.
* unBh mentioned a comparison to baseline methods. We have added a baseline confidence-based prompting method in the appendix.

We encourage the reviewers to respond with any other comments or requests for clarifications that they have - we are happy to discuss any other concerns.

---

### Meta-Review · Area_Chair_SzwD · 2023-12-03

**Metareview:**

The main contribution of the paper is PRIVQA, a multimodal benchmark to evaluate how LLMs and Vision-language models follow instructions about access control. By a set of experiments and red-teaming exercices, the authors show that state-of-the-art models imperfectly meet such privacy requirements and can introduce bias. Even if self-moderation can improve the results, adversaries can by-pass the protection.

On the positive side, an update of the paper has solved several issues raised by the reviewers (access to a public repository, more comparisons with the state of the art and baseline methods and an effort has been made to better position the paper w.r.t. previous works). The proposed dataset is a collection of 5 existing datasets, but the multi-modality aspect is interesting.

The weaknesses are the weak justification of the chosen metrics and their relationship with the privacy risk, the definition of the threat model that should be more clear and convincing.
The contribution is limited because it consists on a set of evaluations without any new contribution wrt methodology, models or algorithms.

**Justification For Why Not Higher Score:**

Perhaps this article would be better placed in a dataset and benchmark track.

**Justification For Why Not Lower Score:**

N/A

---

### Decision · Program_Chairs · 2024-01-16

Reject